# Unleashing Intrinsic Growth Pathways in Regenerating Peripheral Neurons

**DOI:** 10.3390/ijms232113566

**Published:** 2022-11-05

**Authors:** Trevor Poitras, Douglas W. Zochodne

**Affiliations:** Neuroscience and Mental Health Institute, Division of Neurology, Department of Medicine, University of Alberta, Edmonton, AB T6G 2G3, Canada

**Keywords:** nerve regeneration, axon regeneration, peripheral nerve, growth factors, neuropathy, nerve injury

## Abstract

Common mechanisms of peripheral axon regeneration are recruited following diverse forms of damage to peripheral nerve axons. Whether the injury is traumatic or disease related neuropathy, reconnection of axons to their targets is required to restore function. Supporting peripheral axon regrowth, while not yet available in clinics, might be accomplished from several directions focusing on one or more of the complex stages of regrowth. Direct axon support, with follow on participation of supporting Schwann cells is one approach, emphasized in this review. However alternative approaches might include direct support of Schwann cells that instruct axons to regrow, manipulation of the inflammatory milieu to prevent ongoing bystander axon damage, or use of inflammatory cytokines as growth factors. Axons may be supported by a growing list of growth factors, extending well beyond the classical neurotrophin family. The understanding of growth factor roles continues to expand but their impact experimentally and in humans has faced serious limitations. The downstream signaling pathways that impact neuron growth have been exploited less frequently in regeneration models and rarely in human work, despite their promise and potency. Here we review the major regenerative signaling cascades that are known to influence adult peripheral axon regeneration. Within these pathways there are major checkpoints or roadblocks that normally check unwanted growth, but are an impediment to robust growth after injury. Several molecular roadblocks, overlapping with tumour suppressor systems in oncology, operate at the level of the perikarya. They have impacts on overall neuron plasticity and growth. A second approach targets proteins that largely operate at growth cones. Addressing both sites might offer synergistic benefits to regrowing neurons. This review emphasizes intrinsic aspects of adult peripheral axon regeneration, emphasizing several molecular barriers to regrowth that have been studied in our laboratory.

## 1. Introduction

Despite the prevalence and ubiquity of peripheral nervous system (PNS) injury, there are currently no approved therapies available to enhance the regrowth of damaged axons. Surgical intervention and symptom management remain the keystones for management of patients with nerve damage. Following axotomy, it is estimated that only 10% of the transected axons will cross the injury site and will reunite with their targets [1]. This incomplete recovery after PNS damage greatly restricts functional recovery, leaving patients with lifelong impairments or complications. Not only is there loss of motor function or sensory perception resulting from denervation, but there can also be maladaptive changes in the PNS and the central nervous system (CNS). Some changes induce and maintain chronic neuropathic pain. Diabetic polyneuropathy (DPN) is an example of a very common PNS disorder without specific targeted therapy. Unsurprisingly, with the rising prevalence of diabetes, there is a concurrent growth in the number of patients suffering from DPN, adding burdens to over-extended healthcare systems. As sensory axons retract from their normal targets in these patients, there is loss of sensation in the extremities leading to unrecognized damage and infection, with risks of amputation accompanied by neuropathic pain.

Damage to a peripheral nerve initiates an orchestra of biochemical changes in both the axon and the soma. Detailed changes occurring following injury have been reviewed elsewhere [2,3,4]. Shortly after injury, there is a calcium transient in the distal stump that signals for the dismantling of the axon and myelin distal to injury, ‘Wallerian’ or ‘Wallerian-like’ degeneration. This process entails activation of proteases, dedifferentiation of SCs, and recruitment of bloodborne macrophages to the injury site, enabling removal of growth inhibitory myelin components and the distal axon. Additionally, SCs with their basement membranes assemble themselves into columns called Bands of Büngner distal to the injury. These create important guideposts for regrowing axons by releasing molecules that influence their growth and trajectory.

Proximal to injury, retrograde signals are conveyed to the soma, initiating changes in gene expression through activation of regeneration associated genes (RAGs). RAGs are a collection of specific genes that have been implicated in the regenerative response. RAGs include β-tubulin, growth associated protein 43 (GAP43), activated transcription factor 3 (ATF3), signal transducer and activator of transcription 3 (STAT3), c-Jun, among an extensive and growing list of growth facilitating molecules [5,6,7,8,9,10]. These proteins serve diverse roles, acting as signaling molecules, structural components of the elongating axon, and transcription factors. Their expression fundamentally changes the behavior of adult DRG neurons shifting their focus from transmission of sensory signals to an emphasis on growth and recovery.

It is well established that the PNS regenerates axons more robustly than injured CNS axons, but it is incorrect to assume that the heightened growth capacity enables complete recovery following damage. Functional outcomes following a traumatic injury or neuropathy are disappointing and can be attributed to intrinsic and extrinsic challenges that axons must overcome to facilitate complete recovery. Firstly, the growth rate of PNS axons is estimated to be ~1 mm/day in humans, and unless the injury occurs near the axon’s end target, the distance the axon is required to regrow may be several feet in length [5,11]. A sciatic nerve injury in humans for example, is unlikely to fully recover following injury. Moreover, the growth rate of axons over time declines because the regenerative response after damage is not sustained indefinitely. This is due to loss of regenerative support within the long term denervated distal segment, making the distal stump less supportive of axonal growth. Even if growth signals could be enhanced over the longer term, the regrowing axon is guided by cues from the microenvironment to navigate 3-dimensional space to reunite with targets. Unfortunately, axons often become misdirected, for example motor axons growing down sensory paths, axons entering the incorrect distal nerve branch, or axons regrowing retrogradely in the proximal stump, a contributing factor in neuroma formation.

The challenges briefly outlined here restrict PNS recovery, regardless of the regenerative competency relative to the CNS. This review is focused on several specific molecular regenerative pathways and potential intervention points that might lend themselves to eventual therapeutic trials [Figure 1]. We are interested in understanding how neurons utilize these pathways as a component of their intrinsic response to injury and regrowth. Harnessing growth by directing specific intrinsic cellular pathways may be an effective strategy for improving clinical outcomes and quality of life of patients suffering from nerve injury or neuropathy.

## 2. Addressing Experimental Regeneration

Meaningful and rigorous experimental assays of regeneration in models require careful consideration and can be divided into those assessing behavioural, electrophysiological, structural and molecular outcomes (see reviews [1,2,3,5,10]). Some overall, albeit subjective thoughts on these are briefly described here. Combinations of outcomes is likely preferred and most reported studies are in rodents. However, all of these assays have caveats in their interpretation. Behavioural outcomes include thermal and mechanical assays of sensation and motor tests such as gait analysis, hind paw toe spreading or grip power. These tests require multiple, sometimes serial assessments and carefully controlled conditions that include habituation of animals. While pain is not necessarily a regenerative outcome, its assessment requires additional tools, not reviewed here. Electrophysiological outcomes can focus on multifiber motor and sensory conduction, carried out at physiological near nerve temperature and most commonly in the sciatic nerve. Amplitudes of CMAPs (compound muscle action potentials) and SNAPs (sensory nerve action potentials) correlate with numbers of intact and connected motor and sensory axons respectively. Conduction velocities assess the maturation of regrowing axons and their myelination. Further evaluation may involve use of motor unit analysis or single sensory fiber recordings from distal stimulation. Structural endpoints may include, for example, analysis of myelinated and unmyelinated axons at fixed distances and timepoints from the injury. Transverse semithin sections allow measures of myelinated axon numbers, density, caliber and myelin thickness, all indicative of fiber repopulation and maturation. Iso-osmolar fixation and avoidance of drying or handling artifact are essential, but often neglected steps, in generating high quality thin and semithin-sections for analysis. Unmyelinated fiber analysis requires electron microscopy and a sampling strategy involving the nerve of interest. An alternative structural approach uses immunohistochemical sections, for example longitudinal sections of the nerve beyond the injury site to gauge the distance and extent of new axon growth using labels such as SCG10 or others. These are most useful for earlier time points when regrowing fiber density is not extensive. Similarly, analysis of epidermal reinnervation of the skin by labelling axons with PGP 9.5 antibody, may correlate with behavioural sensory recovery. An important caveat in histological studies is that individual profiles may arise from sprouting from a smaller number of parent axons. Retrograde labelling can help to avoid this problem by identifying populations of perikarya that have sent axons into the regenerative zone. Molecular approaches analyze the timing and resolution of neuron markers of injury, including RAGs discussed above. These can also be expressed in outgrowing axons whereas some axon proteins, such as neurofilament, require a degradation time gap before labelled profiles are assumed to be new axons. Finally, the duration, severity and site of injury are important variables that can address whether a candidate approach provides longer-term benefits and is of benefit under more challenging conditions. Studies in rats and larger animals require longer durations to assess meaningful recovery, a feature that makes smaller mouse models easier to work with. Overall, when considering new molecular approaches, in vitro studies of adult neuron outgrowth can predict candidates worth further testing within in vivo models. Studies in non-mammalian systems may offer similar predictions, although specific protein ensembles may differ and are further distant from human interventions. These are not reviewed here.

## 3. Limitations in Extrinsic Growth Factor Support of Regenerating Axons

A large and expanding series of growth factors (GFs) support axon regeneration [2,3,10]. This Chapter will not review these extrinsic determinants of growth in detail. They include the classical neurotrophin family heralded by NGF (nerve growth factor) but also including BDNF (brain-derived neurotrophic factor), NT3 (neurotrophin 3), and Neurotrophin 4/5. This original family has target specificities determined by the expression of their receptors, TrkA for NGF, TrB for BDNF, TrkC for NT3 and TrkB for NT4/5, all of which have varying promiscuity in their actions (Trk = tropomyosin receptor kinase). Their overall roles and the complexity of their interactions continue to be further unravelled and a summary of their signaling is given in Figure 2. Non-neurotrophin growth factors are responsible for a large increase in the numbers of candidates capable of influencing neuron plasticity. Briefly considered, they include the CNTF (ciliary neurotrophic family) with complex receptors that overlap with the cytokines leukemia inhibitory factor (LIF), interleukin 6, granulocyte colony-stimulating factor and oncostatin. CNTF ligates receptor complexes that include gp130, LIFRβ and CNTFRα. Insulin acts as a neuronal growth factor independent of its glycemic actions and operates on insulin receptors (IR) identified on peripheral and central neurons. Insulin and IRs have overlapping ligation and signaling with IGF-1 (insulin-like growth factor-1) and IGF-2 and their receptors. GDNF (glial cell line-derived neurotrophic factor) and its relatives include neurturin, artemin and persephin. These growth factors interact with GFRα1–4 receptors each respectively complexed with RET (Rearranged during Transformation) tyrosine kinase receptors. Growth factors primarily identified with other tissues also have major impacts on neuronal plasticity including the FGFs (fibroblast growth factors) of which there are 23 members now, epidermal growth factor (EGF), hepatocyte growth factor (HGF), platelet-derived growth factor (PDGF), transforming growth factor β (TGFβ), bone morphogenetic proteins (BMPs) that include 14 isoforms, and erythropoietin (EPO). Additional growth proteins are midkine (MK) and pleiotrophin (PTN), osteopontin and clusterin, and the vascular growth factors that include VEGF (vascular endothelial growth factor) and angiopoietins [12,13,14,15]. This long list does not include additional putative, but less explored extracellular growth factors.

Growth factor application in human clinical trials of neurodegenerative conditions and nerve damage or disease has had mixed success. In diabetic polyneuropathy recombinant human (rh) NGF, BDNF and NT3 have had limited benefits. Uncertainties in where and how long to deliver and what dose to use have impacted clinical efforts to translate their use. Side effect profiles are uncertain. For example, intramuscular injections of rhNGF caused local pain. It is uncertain whether GFs should be offered systemically, or locally and whether to deliver them with a carrier or embedded in a scaffold across nerve injury zones. Limitations of growth factor delivery have been an important limitation, summarized in a recent review [16]. These difficulties have arisen from variations in their size, charge, hydrophilicity, stability, and less favourable pharmacokinetic properties such as short half-life, rapid inactivation, rapid clearance, limited permeability, protein binding and potential immunogenicity. Further difficulties may be noise from thermal fluctuations that influence ligand-receptor association, and receptor saturation [17]. Breakdown of the blood nerve barrier and greater permeability of the blood ganglion barrier may allow access of growth factors to peripheral nerves but limitations within sub-compartments may also be limiting and contribute to off target actions. For example, diffusion to and within axons in the endoneurium may be uneven and blood vessels, macrophages, or Schwann cells (SCs) might be inadvertent targets.

The specific impacts of GFs on peripheral nerve regeneration have been reviewed elsewhere [10,18,19]. A theoretical problem with growth factor supplementation after axon damage is the possibility that they may signal a downturn in RAGs that are required to provide intrinsic impetus to regeneration. During regeneration, specific GFs are sequentially and selectively expressed by SCs along a specific time course. This influences what type of motor or sensory axon they partner with. GDNF and PTN in SCs peak at 15 days following nerve injury and preferentially support motor neurons, helping to send them in the correct destination. NGF, BDNF, and IGF-1 are upregulated in sensory SCs within sensory branches with concentrations peaking at about 15 days [20,21,22,23]. Overall, ventral motor roots express PTN, VEGF-1 and IGF-1 whereas sensory nerves express BDNF, NT-3, HGF and GDNF. The largely extracellular glycoproteins osteopontin and clusterin, have differential expression in motor and sensory axon pathways respectively, thus serving as extracellular directional cues. In grafted nerves, osteopontin deletion impaired motor axon regrowth but deletion of clusterin attenuated sensory reinnervation [24].

Despite the evidence for specificity, NGF has also been linked to both motor and sensory axon regrowth [25,26]. Several GFs, specifically NGF, CNTF, insulin, IGF-1 and FGFs 1 and 2 have improved recovery of axons bridging a sciatic nerve transection [27,28,29,30]. To span transections of nerves, specific conduits containing GFs or alternatively bioartificial grafts with embedded GFs have been constructed and tested [18,31,32]. Growth factor gradients within conduits are also important to establish, in order to avoid stalling axons in zones of GF abundance, the ‘candy store’ effect [33]. NGF, BDNF, NT-3, HGF and VEGF have all been applied within regeneration conduits, recently reviewed [34]. Further approaches have applied FGF-2, VEGF, FGF21, bFGF, administered in a variety of ways, including direct injection into silicone tubes, matrigel filled tubes, hydrogel, magnetic nanoparticles, and hydron pellets among others [12,14,35,36,37,38,39,40,41]. GFs have also been delivered directly or using plasmids, adenoviral or lentiviral constructs. VEGF and BDNF mimicking peptides have also been developed [42].

## 4. Summary of Major Intrinsic Growth Pathways of Peripheral Neurons

Downstream of growth factors, several intrinsic pathways that facilitate growth within neurons have been recognized, providing an interesting series of targets that might be manipulated to foster growth. The major, currently recognized intrinsic pathways that support regenerative growth of adult neurons are included here. Schemes summarizing several of the pathways are given in Figure 2 and Figure 3.

### 4.1. PI3K/pAkt (Phosphoinositide 3-Kinase/Protein Kinase B)

Neurotrophic molecules bind to their specific Trk receptors, and this signal is transduced through the membrane, initiating intracellular signaling cascades including the PI3K pathway. The Trk receptors were originally classified as oncogenes with novel receptor tyrosine kinase (RTK) activity [43,44]. Binding of neurotrophins or other growth stimulating molecules to their complementary receptors results in autophosphorylation of specific tyrosine residues within the intracellular domain of the receptor, creating docking sites for proteins containing phosphotyrosine-binding (PTB) or Src homology 2 (SH2) domains [43]. The phosphoinositide 3 kinase (PI3K) complex is a dimer composed of a catalytic and regulatory subunit, p110α(PIK3CA) and p85α(PIK3R1) respectively [45,46]. The regulatory subunit contains a SH2 domain, allowing its association with activated RTKs including the Trk family [45]. This permits the catalytic subunit to generate phosphatidylinositol (3,4,5) triphosphate (PIP3) through phosphorylation of phosphatidylinositol (4,5)-bisphosphate (PIP2) [2,47,48]. The catalytic conversion of PIP2 to PIP3 is necessary for activation of phosphoinositide dependent kinase (PDK), which subsequently phosphorylates Akt, also known as protein kinase B (PKB). Activated Akt then phosphorylates its targets, overall favoring increased cell survival, growth, proliferation and metabolism. Excessive activation of this pathway can result in tumor formation, and thus intrinsic regulation of this pathway is necessary to prevent oncogenesis. One such mechanism is Phosphatase and tensin homolog (PTEN) which dephosphorylates PIP3, consequently resulting in downstream inhibition of pAkt action [2,45,49] (Figure 1).

#### 4.1.1. PI3K/Akt Regulation of Apoptotic Cascades

During regeneration, loss of parent neurons reduces the population of unique axons available to connect to target tissues. While controversial, retrograde apoptosis in adult peripheral neurons after axotomy may contribute to this outcome. However, while not articulated in the literature, stable postmitotic neurons might exploit anti-apoptotic pathways and redirect them to support growth when survival is not at stake. Thus, stimulation of PI3K and subsequent Akt activation has been recognized to be of critical importance in regulating cell survival. Phospho-Akt exhibits influence over apoptotic pathways by blocking proapoptotic signals while simultaneously stimulating pathways involved in cell survival. For example, the Bcl-2 family member Bcl-2 associated agonist of cell death (Bad) is a proapoptotic protein that binds to anti-apoptotic Bcl-2 proteins, thereby shifting the balance of pro- and anti-apoptotic signaling towards cell death [50]. Active phospho-Akt (p-Akt) influences this balance by phosphorylating Bad on ser136 allowing it to complex with 14-3-3 proteins in the cytoplasm, thus preventing its inhibition of the anti-apoptotic Bcl-2 family members, Bcl-2 and Bcl-XL [51,52,53]. Moreover, apoptotic pathways converge to activate caspases which are proteases that breakdown cellular components enabling systematic and organized cellular destruction. Caspase-9 is an initiator caspase that when released from its pro-domain by proteolytic cleavage, can activate other caspases like caspase 3 and 7 to facilitate apoptosis. p-Akt has been reported to directly phosphorylate caspase 9 on serine 196 rendering it inactive [54]. Akt influences apoptosis through these cascades by post translational modifications, however there are transcriptional mechanisms involving Akt that can impact survival. For instance, forkhead box O (FOXO) is a transcription factor that promotes transcription of proapoptotic genes like the BH-3 containing members of the Bcl-2 family. FOXO is a substrate for Akt phosphorylation, and like Akt’s control over Bad, this phosphorylation allows 14-3-3 proteins to bind to FOXO, displacing it from its target genes. This renders FOXO unable to initiate transcription of pro-apoptotic genes. Akt further elicits control over the survival of cells through interaction with other molecules like Mdm2, IKK, and CREB to name a few (reviewed here) [55].

#### 4.1.2. PI3K/Akt Effector mTOR

The PI3K/Akt pathway further influences cellular behavior by activation of critical cellular growth mechanisms unrelated to apoptosis. This is in part achieved by activation of the mammalian target of rapamycin complexes (mTORC1/2), which are critical nutrient sensors and mediators of growth factor signaling [49]. Downstream activation of mTORC1 is achieved by Akt-mediated phosphorylation of tuberous sclerosis complex (TSC1/2) leading to its inactivation and inability to regulate Ras homologue enriched in brain (Rheb) [49,55,56]. Rheb then subsequently activates mTORC, enabling interaction with its effectors that are involved in protein translation, ribosomal biogenesis, and metabolism. Activation of the mTORC complex is often accompanied by increases in cellular growth, exemplifying its therapeutic potential in the context of axon regeneration. For example, it was recently discovered that mTOR becomes locally translated in injured peripheral axons and deletion of the mTOR’s 3′ untranslated region (mTOR 3′UTR) caused a reduction in the total amount of local protein production [57]. In agreement with this, blocking mTOR activity peripherally at the axon decreases protein production and growth cone formation independent from processes occurring at the cell body [58]. It has been demonstrated that stimulation of mTOR after injury can also be observed as a component of the regenerative response within the DRG [59]. In the same study, deletion of the negative mTOR regulator TSC2 amplifies mTOR activation and subsequently favors superior regrowth. Conversely, genetic ablation or inhibition of mTOR after injury in mice greatly suppresses axon elongation and growth cone formation [58,60]. The growth effects of mTOR activation are not exclusive to peripheral axons, within the CNS increased mTOR activation is associated with greater retinal ganglion cell regeneration after injury, while its inhibition impairs their growth capacity [61].

Despite this evidence, the role of mTOR in PNS recovery following damage is controversial. A study from our laboratory demonstrated that increasing the activity of PI3K/Akt after injury through PTEN inhibition drives regeneration in the presence of the mTOR inhibitor rapamycin [62]. Furthermore, mTOR hyperactivation can be observed in diabetic patients with small fibre neuropathy, a condition where it can be argued that a regenerative deficit is present [63]. In support of these findings, administration of rapamycin attenuated hyperalgesia in a model of experimental type1 diabetes [63,64]. Based on these studies, it is difficult to conclude whether active mTOR is compulsory for driving more robust regeneration following axonal damage. However, it is understood PI3K/Akt activation are associated with enhanced regeneration.

#### 4.1.3. PI3K/Akt Effector GSK3β (Glycogen Synthase Kinase 3)

GSK3β is widely expressed in all tissues and in the absence of active Akt, GSK3β is constitutively active. Upon phosphorylation by Akt on serine9, GSK3β loses its catalytic activity, silencing its impact on downstream targets. Originally discovered as a regulator of glycogen metabolism, GSK3β has since been implicated in neural development as well as pathological processes associated with chronic diseases including diabetes, neurodegeneration, and cancer (see reviews [65,66,67,68]). Despite extensive study, literature surrounding the role of GSK3β remains unclear. For example, one study found that when phosphorylation resistant mutants of GSK3 (GSK3α-S21A/GSKβ-S9A) were knocked-in within axons, they display accelerated growth properties [69]. It is therefore argued by some groups that sustained GSK3 signaling is required for axon regrowth. In contrast to this, other groups have found that suppressing GSK3 facilitates more favorable regenerative outcomes. For instance, a study by Jiang et, al demonstrated that GSK3β expression is regulated by the microRNA miR-26a [70]. Increases in miR-26a were associated with reduced GSK3β. Conversely, reducing miR-26a through treatment with antisense oligonucleotide sequences (a.k.a antimiR) impaired miR-26a dependent suppression of GSK3β, ultimately yielding less neurite outgrowth [70]. In support of these findings, it was also discovered that the GSK3 response in DRG neurons depends on which axonal branch is damaged. Recall that DRG neurons are pseudo-unipolar, referring to the single axon that leaves the cell body and bifurcates into a central and peripheral branch. The central branch enters the spinal cord to interact with second order neurons and convey information to the brain, while the peripheral branch extends toward their sensory targets. In the peripheral, but not the central branch, injury results in an intrinsic reduction in GSK3β transcription [71]. It can be speculated that the central branch’s inability to decrease GSK3β is related to its poor regenerative capacity.

Adding further complexity, GSK3 has been also observed to play an additional, pleiotropic role within growth cones. The growth cone forms post injury and is an exceptionally dynamic structure that changes based on molecular cues received from the microenvironment. There is evidence suggesting the GSK3 is involved in regulating the structure and behavior of growth cones. Along these lines, GSK3 has been observed to accumulate at the leading edges of growth cones mediating their collapse when exposed to repulsive cues like Semaphorin 3a [72]. A separate study found that GSK3 was implicated in promoting axon regrowth by allowing microtubule stabilization and growth cone protrusion [72,73]. Taken together, it appears the effect of GSK3 signaling on regenerating peripheral neurons is context and likely concentration dependent.

### 4.2. Ras/ERK (Rat Sarcoma Virus/Epidermal Growth Factor Receptor Extracellular-Regulated Kinase)

The Ras/ERK pathway is linked to augmented cellular growth, proliferation and regeneration [74,75]. Activated ERK has over 100 known substrates including transcription factors, and thus its influence in the cell cannot be understated [76]. The Ras/ERK pathway begins with extracellular growth factors like neurotrophic molecules binding to their complementary receptors. These receptors can be a receptor tyrosine kinase, or a G-protein coupled receptor [74,77]. Similar to the PI3K pathway, ligand binding to an RTK leads to autophosphorylation of the receptor creating an anchoring point for the adaptor protein Grb2, allowing for subsequent binding of the guanine exchange factor (GEF), Son-of-sevenless (SOS) [78]. As a member of the Ras GTPase super family, Ras requires a GEF to displace GDP bound to the inactive protein to facilitate its activation. Once activating signals are abated, GTPase activating proteins (GAP) will help initiate the intrinsic GTPase activity, rendering Ras inactive [74,79]. Once active however, Ras binds to Raf, enabling Raf’s kinase activity. Raf phosphorylates MEK1/2 resulting in the subsequent phosphorylation of ERK1/2 by phosphorylated MEK. Upon activation, phospho-ERK1/2 is transferred to the nucleus where it can phosphorylate transcription factors including c-Fos, c-Jun, c-Myc, and ATF-2, among other targets. These transcription factors influence the global cellular environment and promote cellular processes like survival, growth and proliferation [74,80].

Given the oncogenic potential of the Ras/ERK pathway, it must be tightly regulated to prevent malignant transformation. There are several regulatory systems that prevent excessive Ras/ERK signaling. One mechanism used by cells to prevent overactivity of this pathway is through a family of molecules known as mitogen kinase phosphatases (MKPs). MKPs directly dephosphorylate MAP kinases, thus preventing downstream transcription factor activity. A subset of these MKPs are transcriptionally regulated by the MAPK cascade, thereby providing feedback inhibition [81]. The dual specificity protein phosphatase family (DUSP) is comprised of 10 (thr/tyr) phosphatases which function as MKPs to limit MAPK cascade signaling [82]. Furthermore, regulation occurs upstream of ERK as well. Another group of proteins that become transcriptionally induced by activated ERK are the Sprouty (SPRY) family of molecules [83]. These proteins regulate the RTK response and therefore prevent downstream activation of ERK. For example, SPRY1 and SPRY2 prevents Ras activation by occupying its binding site on Grb2, preventing Ras recruitment and subsequent activation [75,84]. Similarly, Sprouty-related EVH domain containing protein (SPRED) is another upstream inhibitor of ERK signaling that blocks phosphorylation and activation of Raf [85]. Like many oncogenic pathways, there is redundancy in the regulatory mechanisms that prevent aberrant ERK signaling and these can be targeted to accentuate the repair response. In the context of peripheral nerve injury, the Ras/ERK pathway may be utilized to enhance axon regrowth following damage.

#### 4.2.1. Ras/ERK and the Intrinsic Regenerative Response

Neurotrophic signaling through the Trk receptors is well known to be associated with activation of the Ras/ERK pathway. Moreover, the intrinsic regenerative response to damage includes increased phosphorylation and activation of ERK both in the proximal and distal stump post axotomy [86,87,88]. In support of these findings, facial nerve transection heightened the phosphorylation of ERK with the highest p-ERK:ERK ratio occurring 7 days post injury [89]. With ERK activation occurring in axons, along with the understanding that ERK targets a subset of transcription factors requiring nuclear localization, there is a requirement for retrograde transport to the soma. Following injury, local production of soluble vimentin occurs alongside axoplasmic phosphorylation of ERK [90]. Activated ERK complexes with vimentin, allowing importin-β binding. This complex protects ERK from dephosphorylation and enables the molecular motor protein dynein to transport p-ERK to the nucleus [90,91]. It is therefore clear that part of the axonal response to damage includes local rises in p-ERK, along with vimentin dependent retrograde transport back to the soma, where it can influence neuronal behavior.

Increases in ERK phosphorylation have been investigated for its impact on the ability of neurons to regrow. For example, primary cell cultures derived from the superior cervical ganglia or embryonic DRGs demonstrated that ERK activation was coupled with increased axon elongation, while simultaneous activation of the PI3K pathway was better associated with neuronal survival in addition to axon elongation [75,92,93]. This increase in growth capacity and associated surge in ERK phosphorylation can comparably be observed in adult DRG neurons treated with the neurotrophic molecule, fibroblast growth factor (FGF2) [94,95].

As described above, targets for p-ERK phosphorylation include transcription factors, giving p-ERK influence over diverse cellular processes like cytoskeletal remodeling, metabolism, growth, and proliferation. Recently c-Jun has been found to regulate a subset of RAGs, promoting adoption of a regenerative phenotype [96]. This is not unique to c-Jun, since the proto-oncogene c-Myc has been discovered to provide transcriptional control over RAGs post injury, inducing a transition to a regenerative phenotype [97]. Upstream control of transcription factors like c-Myc and c-Jun gives activated ERK control over wide and diverse cellular processes. This exemplifies the Ras/ERK pathway as a potential therapeutic intervention point for translational therapies aimed at augmenting recovery from nerve injury.

Despite the studies above demonstrating ERK as a positive influence in the regenerative milieu, other studies question the utility of ERK signaling in injury induced regrowth. For example, growth stimulation of cultured DRG neurons through application of NGF is unaffected when combined with the MEK inhibitor PD 98059 [98,99]. Another report found that inhibiting MEK with a separate small-molecule inhibitor, UO126, in combination with IGF-1 and NGF yielded an unexpected increase in neurite branching [100]. However, blocking the PI3K pathway with LY294002 extinguished the growth promoting effects of the NGF and IGF-1 combined therapy [100]. Moreover, regeneration is not only dependent on the neuronal response. Glial cells, namely SCs play a critical role in Wallerian degeneration, regrowth, and guidance. It has been observed that overactivity of the Ras/ERK pathway in SCs through a SC specific conditional knock-in mouse was characterized by morphological abnormalities, reduced intraepidermal fiber density and reduced functional regeneration following nerve transection [101]. Furthermore, ERK activation within both peripheral and central neurons and glia have been implicated in the induction and maintenance of neuropathic pain (reviewed elsewhere) [102,103,104,105].

#### 4.2.2. Targeting the Ras/ERK Pathway

Regardless of the controversy regarding ERK’s role in the intrinsic regenerative response, some approaches to augment ERK activation have been evaluated. An approach toward increasing ERK signaling is to block upstream inhibition that normally limits ERK activation. For example, Sprouty2 knockdown in dissociated DRG cultures using siRNA has revealed an improvement in neurite outgrowth [94]. Sprouty2-deficient mice showed superior functional regeneration with heightened pERK activation with concomitant administration of NGF or FGF [106]. Augmented pERK in Sprouty2 null mice is unsurprising given that Sprouty proteins are classically known to limit the RTK response. Along these lines, it was discovered that Sprouty2 itself is regulated by another kinase, testicular protein kinase 1 (Tesk1) [107]. This novel kinase disrupts the Sprouty2 and Grb2 interaction, enabling Ras/ERK signaling to proceed unimpeded [107]. Furthermore, a synergistic effect was observed using a combined approach to amplify both the PI3K/Akt and Ras/ERK pathways simultaneously [108].

Another potential strategy is to prevent pERK dephosphorylation to allow for more sustained pERK signaling. Our lab recently explored DUSPs as a potential target for regeneration. Surprisingly, DUSP1 and -4 knockdown in DRG cultures revealed a reduction in neurite outgrowth, while in vivo our lab found that inhibition of DUSP accelerates axon degeneration in the nerve distal to axotomy [109]. The impacts of DUSP knockdown may be complex involving ERK, likely context dependent, but may also impact a separate axonal degeneration pathway mediated by SARM1 and overall may be subject to constraints within the cellular environment.

### 4.3. JAK/STAT (Janus Kinase/Signal Transducer and Activator of Transcription)

The JAK/STAT pathway in the context of axonal regeneration has been extensively reviewed elsewhere [110,111,112,113]. In brief, the signaling pathway begins with cytokines binding to their complementary receptor, leading to receptor subunit dimerization. JAK then associates with the intracellular domain of the receptor allowing for transphosphorylation and activation of JAK. Activated JAK phosphorylates tyrosine residues on the intracellular domain of the receptor, creating docking sites for SH2 domain containing proteins like the STAT family. The SH2-phosphotyrosine interaction anchors STAT in close proximity to activated JAK, resulting in phosphorylation of tyrosine residues on STAT. Phosphorylated STAT disassociates from the receptor, and dimerizes through interactions between the phosphotyrosine of one STAT monomer with the SH2 domain of the other. STAT Homo- or heterodimers then migrate to the nucleus where they can stimulate transcription directly or through interactions with other non-STAT transcription factors. Both JAK and STAT have multiple isoforms, with 4 identified JAK proteins (JAK1, JAK2, JAK3, tyrosine kinase 1) while there are 7 members of the STAT family. However, STAT3 has been most heavily implicated in the neuronal response to damage and therefore will be our primary focus.

The JAK/STAT pathway can be positively regulated by other proteins specific to the receptor activated [110]. Conversely, there are also several regulatory mechanisms to ensure that JAK-STAT signaling does not occur unfettered. Firstly, suppressor of cytokine signaling (SOCS) and cytokine inducible SH2 containing proteins (CIS) belong to a family of structurally related proteins that are induced by STAT transactivation that provide feedback inhibition on JAK/STAT signaling [110,114]. SOCS/CIS limits JAK/STAT signaling through multiple distinct mechanisms that include: preventing STAT recruitment to the receptor by binding phosphotyrosine residues, blocking JAK’s kinase activity through direct binding of JAK, or through association with other proteins resulting in polyubiquitination and proteasomal degradation of JAK and STAT proteins [110,114,115,116,117]. Secondly, protein inhibitor of activated STAT (PIAS) can block STAT interaction with DNA, or can form complexes with other coregulatory proteins to prevent DNA binding and transcription [110]. Lastly, protein tyrosine phosphatases (PTPs) dampen JAK/STAT as a result of dephosphorylation of tyrosine residues on activated JAK or on the receptor itself [110].

#### JAK STAT Signaling and Regeneration

Unlike the signaling cascades discussed above, the JAK/STAT signaling axis is not activated by classical neurotrophic Trk receptors. Neuropoietic cytokines such as ciliary neurotrophic factor (CNTF), interleukin 6 (IL-6), and leukemia inhibitory factor (LIF) initiate JAK-STAT signaling by binding to their receptor. Cytokine receptors differ in their subunit composition depending on their ligand specificity, however, they do all share a common gp130 subunit that also contributes to substrate specificity. Similar to the other pathways discussed above, the JAK/STAT pathway becomes activated following axotomy as part of the intrinsic regenerative programing [88,118,119,120]. Damage to axons results in release of cytokines like LIF and IL-6, both of which converge on STAT3 activation, further bolstering the repair response [121,122]. Upregulation of STAT3 is a common finding within sensory, motor and autonomic neurons following axotomy or damage [120,123,124]. For example, sciatic nerve transection initiates STAT3 signaling with levels of STAT3 peaking 6 h post injury, and heightened active STAT3 expression remains detectable 1 month following injury [118]. This phenomenon is similarly observable following neurotoxic application of capsaicin to axons, where heightened p-STAT3 was measured several weeks post capsaicin treatment [120]. Furthermore, 3 fold-lower levels of p-STAT3 expression can be detected in long-term STZ induced type 1 diabetic mice, likely contributing to the poor regenerative capacity observed in diabetic neurons [125]. In support of this, diabetic animals treated with CNTF, had a corresponding increase in STAT3 phosphorylation and partial rescue of diabetic neuron growth deficiency [125].

To further exemplify the essential role of STAT3 in axon regrowth post injury, studies have been performed showing loss of regenerative competency in neurons if JAK/STAT3 is inhibited or genetically knocked out. Sensory neuron specific deletion of the gp130 subunit nullifies neuropoietic signaling, reducing downstream STAT3 phosphorylation [126]. Consequently, these conditional knockout animals demonstrated impaired functional recovery and skin innervation post nerve crush [126]. Furthermore, axotomy performed in animals with STAT3 ablated are similarly characterized by regenerative deficits, particularly initiation of regrowth programs [8]. Conversely, in the same study, STAT3 overexpression in DRG neurons resulted in a 400% increase in collateral sprouting of the DRG central branch following a dorsal column injury [8]. STAT3 signaling is not unique to sensory neurons and within motor neurons, STAT3 functions as a critical survival factor. Motor neuron specific ablation of STAT3 drastically reduced the number of motor neurons within injured facial nuclei, further supporting the hypothesis that STAT3 serves an indispensable role in the regenerative milieu [123].

Within the more growth restricted CNS, the JAK/STAT pathway has been similarly evaluated. One study found that the transcription factor KLF4 interacts with p-STAT3 to limit its ability to regulate gene expression [127]. Additionally, in this study they found that deletion of KLF4 drives superior and more robust axon regeneration following an optic nerve crush. However, enhanced growth was absent if STAT3 was deleted in combination with KLF4 [127]. In support of these findings, through deletion of the cytokines LIF and CNTF, it was determined that these cytokines and downstream activation of STAT3 are essential for the growth mediated by a conditioning lens injury or by intravitreal injection of the inflammatory molecule zymosan [128]. There are a several studies demonstrating growth stimulatory and neuroprotective benefits of activated STAT3 in the central nervous system [127,129,130,131].

JAK/STAT pathway activation as a component of the response to damage is not unique to the neuronal population, but also has importance in Schwann cells. Despite reports that STAT3 is dispensable during SC development, STAT3 has been discovered to be essential in the maintenance of the SC repair phenotype [132]. Therefore, as SCs dedifferentiate and become more repair oriented, STAT3 expression should presumably be upregulated after injury. STAT3 upregulation in SCs was experimentally achieved through expression of the long non-coding RNA (LncRNA) SNHG16 following a sciatic nerve injury [133]. SNHG16 ‘sponges up’ miR-93-5p, a microRNA that that targets STAT3 mRNA (among other targets) for destruction [133]. However, STAT3 activation post injury does not simply regulate growth, it can also participate in maladaptive changes. For example, SCs release CNTF following injury that acts on sensory neurons, increasing STAT3 activation. CNTF action on sensory neurons induce IL-6 expression, neuroinflammation and consequently, neuropathic pain [134]. Moreover, diabetic SCs exhibit a deficiency in autophagy, a process that is important in the clearance of myelin and axon debris following injury. The reduction in autophagy is partly dependent on STAT3 phosphorylation as application of AG490, a Jak/STAT inhibitor yielded partial rescue of autophagy [135]. Taken together, these studies provide extensive evidence that STAT3 signaling is complex and governs diverse biological functions during PNS regeneration.

### 4.4. Wnt/β-Catenin Signaling

The Wnt/β-catenin pathway is another indispensable and complex signal transduction pathway that is important in development, normal biological function, disease, and regeneration. This pathway has been reviewed in detail elsewhere (see reviews [136,137,138,139]). In the resting state when this pathway is not stimulated by Wnt ligands, a complex of proteins called the destruction complex bind to and phosphorylate β-catenin, targeting it for ubiquitination and proteasomal degradation. The destruction complex is composed of adenomatous polyposis coli (APC), GSK3β, Axin1, and Casein kinase 1 (Ck1). However, when Wnt molecules bind the Frizzled receptor on the cell surface, it recruits the co-receptor low density lipoprotein receptor related protein 5 and 6 (LRP5/6). LRP5/6 become subsequently phosphorylated, and this causes dishevelled (DVL) to be recruited. DVL inhibits the destruction complex, releasing β-catenin, enabling its translocation to the nucleus. Once β-catenin is in the nucleus, it binds to T cell activator (TCF)/Lymphoid enhancer binding factor (LEF) and other coactivators to facilitate transactivation of β-catenin target genes.

Adding to the complexity, there is an independent non-canonical pathway associated with Wnt/β-catenin signaling. In short, Wnt ligands bind to FZD receptors and co-receptors ROR1/ROR2/Receptor tyrosine kinase (Ryk) are recruited and activated. Depending on the coreceptor, as well as cellular environment, non canonical Wnt/β-catenin signaling can cause cytoskeletal remodeling through Rac1/RhoA, release of Ca^2+^ from the endoplasmic reticulum, as well as activation of growth and survival pathways through PI3K/Akt crosstalk [139,140].

Compared to the other growth pathways discussed above which responded to axonal damage by increasing activity, a counter intuitive reduction in β-catenin expression is observed. Our laboratory found that injury to peripheral nerves induces lower levels of β-catenin while paradoxically upregulating APC, a member of the β-catenin destruction complex [141]. Another research group confirmed these findings, observing limited β-catenin expression after injury [142]. Specifically, in this study, it was determined that deletion of Galectin-3 (gal-3), an important binding partner and regulator of β-catenin, corrected the decrease in β-catenin [142,143]. Consequently, in vivo regeneration was more extensive and was accompanied by an improvement in sensory and motor behaviour in gal-3 knockout animals [142]. These studies support the concept that Wnt/β-catenin is perturbed following PNS injury and if β-catenin levels are restored, superior functional regeneration ensues.

Wnt/β-catenin signaling accentuates axonal regeneration beyond the PNS, and similar effects are seen within models of CNS injury. For example, cultured retinal ganglion cells (RGCs) display enhanced growth properties when the Wnt ligand Wnt3a was included in the media [144]. In support of these findings, in vivo application of Wnt3a after optic nerve crush increased activity of β-catenin, coinciding with improved regeneration [145]. These findings are supported by identifying that Wnt3a application to cultured spinal cord neurons triggered greater neurite outgrowth [146]. Several studies have shown that downstream activation of Wnt/β-catenin signaling offers a therapeutic benefit within different animal models of spinal cord injury [147,148,149]. Despite the axonal benefits of the canonical Wnt/β-catenin pathway, activation within glial cells can accelerate scar formation, therefore reducing axonal growth [150].

It should be noted that not all Wnt ligands are created equal, and some have been found to serve as repulsive growth cues and inhibitors of neurite outgrowth [151]. Generally, signaling through the canonical pathway is associated with augmented growth, while non-canonical activation of the Wnt/Ryk pathway provides an obstacle to regeneration [151,152,153,154,155]. Specifically, blockage of Wnt signaling by endogenous expression of Wnt inhibitory factor 1 (WIF1) or secreted frizzle related protein (SFRP) in preconditioned sensory neurons improved their regenerative capacity within the normally growth restricted central branch [151]. A similar phenomenon was observed when exogenous WIF1 or SFRP2 was administered in DRG explants and within a sciatic crush model of nerve injury [152]. Non-canonical Wnt signaling post injury can additionally foster maladaptive changes associated with chronic pain syndromes commonly seen post nerve damage [156,157]. The non-canonical Wnt/β-catenin signaling cascade also has implications within the CNS. Wnt molecules are found to be upregulated following a dorsal hemisection of the spinal cord that restricts the recovery of descending corticospinal tract axons [153].

The duality of the Wnt/β-catenin network enables diverse cellular impacts depending on the Wnt ligand, as well as the co-receptors recruited. The canonical and non-canonical Wnt pathways have different downstream effectors and as such, they modify cellular behaviour in distinct ways. Within the regenerating PNS, the overall consensus is that selective activation of the canonical pathway enhances nerve regeneration. Conversely, non-canonical Wnt signaling is frequently coupled to decreased axonal growth.

## 5. Regenerative Brakes and Their Manipulation

Adult neurons, including those of the PNS, are regeneration ‘reluctant’, a status that may be important in maintaining the integrity and fidelity of the carefully wired nervous system. However, following injury, regeneration ‘reluctance’ is an impediment to regrowth. While CNS neurons may also be inherently less ‘plastic’ or willing to regrow, they do have the capacity to enter and traverse peripheral nerve bridges, classical work by Richardson and colleagues [158]. Much more of the literature invokes extrinsic mechanisms, for example NOGO signaling to account for the severe regenerative deficit of CNS neurons [159,160]. In the PNS, limitations in regeneration are evident and common in humans who have undergone traumatic nerve injury or axonal damage from neuropathy. Despite their capacity for growth, PNS neurons fail to reinnervate distal targets if pathways for regrowth are disrupted, perhaps by transection or if there are long distances involved to reconnect [1]. While there are undoubtedly extrinsic barriers to more successful regrowth in the PNS, regeneration ‘reluctance’ is readily demonstrated by the overall slow launch of sufficient populations of new axons to reach distal nerve segments or targets before those environments become unfavourable to growth. Hesitant growth can be linked to intrinsic molecular ‘roadblocks’ that are constitutively expressed or change during injury within neurons. RAGs expressed after an axotomy injury of nerve, have well explored relationships to regeneration, with several ‘hits’ identified that influence neuron plasticity or regrowth. However there has been an assumption that most if not all growth determinants can be identified on RAG screens, for example using RNA sequencing [161,162]. While powerful, these tools have generated large numbers of upregulated or downregulated mRNAs that rely on imperfect gene ontology (GO) analysis to classify their roles. Intuition is required to sort out which targets may be worth exploiting, a task not helped by GO; many molecules can fit into one to several categories of cell biology. Some of the molecular brakes on regeneration may not undergo differential regulation following injury but retain a constitutive inhibitory function, with or without injury.

Preconditioning, the enhanced plasticity and growth behaviour of peripheral neurons after a prior distal axon injury, or axotomy, is often considered a goal for enhancing regeneration [163]. However, it is not a clinically applicable strategy to improve nerve regrowth after trauma or neuropathy whereas mimicking its benefits has been an aim in regeneration studies. For example, exogenous electrical stimulation, of benefit in models and early human studies, may ramp up RAGs to accomplish this goal [13,164,165,166]. Its impact is considered in more depth in other chapters of this issue. However, enhancing neuron plasticity to the levels of preconditioning may be limiting. Manipulation of intrinsic neuron pathways that attenuate or block regeneration may allow levels of plasticity that substantially exceed those offered by preconditioning or RAG upregulation. Use of both strategies may be synergistic and necessary to enable functional recovery post injury.

Roadblocks to regeneration, with a focus on peripheral neurons, can include those expressed in the perikaryon cytoplasm or nucleus, or those expressed distally in growth cones. In the latter case, local axonal synthesis may be important in their expression and action [167], another mechanism not captured by a focus on RAG expression in parent cell nuclei. Local expression however may also drive retrograde signaling to the perikaryon and nucleus, thereby influencing the behaviour of the full neuronal tree [168]. Some roadblocks, such as PTEN, can be closely linked to well characterized intrinsic growth programs, as discussed. The exact role or fit of others, however remains to be clarified. Several are transcription factors, but whether their repertoire of activated transcripts are completely unique or overlapping is unknown. Several of the regenerative roadblocks have been identified from the tumour biology literature, understood as tumour suppressor molecules or proteins that interact with and brake oncogene activation. While these proteins are best understood in tumour cells, where they participate in unrestrained cellular growth, they are not unique to cancer and their expression in stable, nonplastic neurons has been surprising. In addition, manipulation of these key molecules in neurons should not dissuade later translational ideas over their potential clinical use. Oncogenesis is a long term multistep process, not likely to be initiated by local targeted approaches in the nervous system over constrained timetables. For example, work on the Retinoblastoma 1 protein, considered below has previously identified a specific need for more than one ‘hit’ to trigger tumorogenesis. Moreover, understanding the full interactome of tumour suppressor proteins in neurons may allow recognition of associated proteins that impact neuron growth but have not had known links to oncogenesis. Finally, several of the current approaches toward enhancing growth using GFs, including insulin, also possess similar low theoretical risks of impacting tumour growth. With these caveats in mind we outline experience with four specific roadblocks to regeneration and the evidence for their roles in regrowth.

### 5.1. PTEN (Phosphatase and Tensin Homolog Deleted on Chromosome Ten)

PTEN is a tumour suppressor phosphatase molecule that inhibits the PI3K-pAkt intrinsic growth pathway, specifically by acting to shift the downstream PIP3 (phosphatidylinositol 3,4,5 triphosphate) back to PIP2 (phosphatidylinositol 4,5 diphosphate). Mutations of PTEN have been identified in carcinomas and in Cowden’s syndrome, a condition associated with an increased risk of oncogenesis. PTEN mRNA and protein are expressed in the cytoplasm of DRG sensory neurons, and in Schwann cells (SCs) [62]. Whereas overall PTEN expression declined at 3d after injury, its phosphorylated version, with marked nuclear expression did decline. PTEN had an unexpected prominent and intense expression in small caliber sensory non-peptidergic neurons (isolectin-B4-binding), independently identified as slower growing [169]. The selective PTEN inhibitor dipotassium bisperoxo(pyridine-2-carboxyl) oxovanadate [bpV(pic)] generated a dose dependent rise in adult sensory neurite outgrowth and rises in pAkt. Using siRNA directed to PTEN, we confirmed similar rises in outgrowth. Moreover, outgrowth was yet more pronounced in preconditioned neurons (3 days earlier) indicating an additive impact beyond that offered by preconditioning. Despite the assumption that PTEN suppression increased neurite outgrowth by activating mTOR, rapamycin, an inhibitor of mTOR had no impact on neurite outgrowth. To test the impact of PTEN KD in vivo, we applied the inhibitor or siRNAs through a subcutaneous access port [170] within regeneration conduits that spanned the 3–5 mm gap across the transected sciatic nerves of rats and examined axon outgrowth at 7 days. Both the PTEN inhibitor and siRNA were associated with rises in axon outgrowth from the proximal stump into the tissue bridge spanning the conduit. Similar impacts have been confirmed, for example by identifying that that neurite outgrowth is accentuated by miR-222 which targets PTEN transcripts for destruction [171]. Enhanced growth from PTEN suppression also parallels concurrent CNS work in optic nerve regrowth, albeit differing in its relationship to mTOR involvement [61]. Overall, the findings indicate that PTEN, despite a fall in its expression after injury, nonetheless operates as a regenerative roadblock in both naïve and preconditioned neurons.

PTEN manipulation appears to influence regeneration when preferentially targeted at the perikaryal level, an impact that might then ramify throughout the neuron. For example, a PTEN inhibitor directed to growth cones in a turning assay had little impact. In contrast, similar gradients instead directed to the perikarya induced growth of multiple neurites including those at some distance to the perikarya, indicating participation of the full neuron tree [172] (Figure 4). However, PTEN is also expressed in distal axons and growth cones, and it is likely that local impacts have not been fully explored.

Nedd4 (neural precursor cell-expressed developmentally down-regulated protein 4), is a ubiquitin protease system (UPS) E3 ligase that is colocalized with PTEN in sensory neurons and degrades it. In adult sensory neurons [173] and in Xenopus retinal ganglion cells [174] Nedd4 impacted outgrowth through its impact on PTEN. For example, Global UPS suppression reduced neuron growth whilst preserving or upregulating PTEN expression and specific knockdown of Nedd4 increased PTEN and thereby attenuated neurite outgrowth. These findings identify an endogenous mechanism of downregulating PTEN and highlighted the role of the UPS in supporting neuron regrowth.

### 5.2. Rb1 (Retinoblastoma 1)

Rb1 is a tumour suppressor protein, also classified as a ‘pocket protein’ that inhibits of cell cycle progression from the G1 phase to the S phase by binding E2F transcription proteins. Mutations of human Rb1 from homozygous mutation or loss of heterozygosity are associated with the development of retinoblastoma. Phosphorylation of Rb1 inhibits its binding to E2F, inactivating its transcriptional repression. Given the divergent transcriptional signaling by E2F, we hypothesized that Rb1 KD might support growth and plasticity of non-replicating neurons [175].

Rb1 is expressed widely in the cytoplasm, nuclei and axons of large and small caliber DRG neurons and undergoes only modest declines in expression after axotomy. SC expression is limited. Dissociated and preconditioned adult rat or mouse sensory neurons undergoing siRNA KD had dramatic rises (50%) in neurite outgrowth, maximum process length and branching (Figure 5). Moreover transfected DRG neurons from adult Rb1-floxed mice (FVB; 129-Rb1tm2Brn) with control adenovirus (Ad-GFP (green fluorescent protein)) or Cre recombinase expressing adenovirus (Ad-CMV (cytomegalovirus)-Cre) similarly generated substantial rises in neurite outgrowth. There was no evidence of neuronal apoptosis using assays of capsase-3 activation, DNA damage or LDH release from Rb1 KD and downstream pAkt, Pak1 or CDK5, some specific mediators of plasticity, were not altered. However neuronal PPARγ was upregulated and a PPARγ agonist increased outgrowth. An antagonist of PPARγ attenuated the impact of Rb1 KD and suppressed adult neuron plasticity. While PPARγ may be a key downstream growth mediator of Rb1 KD, its impact was less robust than that of Rb1 KD.

Rb1 siRNA applied within transected nerve conduits, as described above, was associated with local Rb1 mRNA KD, greater and longer axon growth, and more outgrowth of accompanying SCs by day 7 after injury. Given the close axon-SC partnership in regrowing nerves, greater axon regrowth during this intervention supported active SC participation [176]. In mice with sciatic nerve crush, local ipsilateral DRG Rb1 siRNA KD was associated with improvements in mechanical sensation, thermal sensation, hindpaw grip strength and sensory conduction velocity.

**Figure 5 ijms-23-13566-f005:**
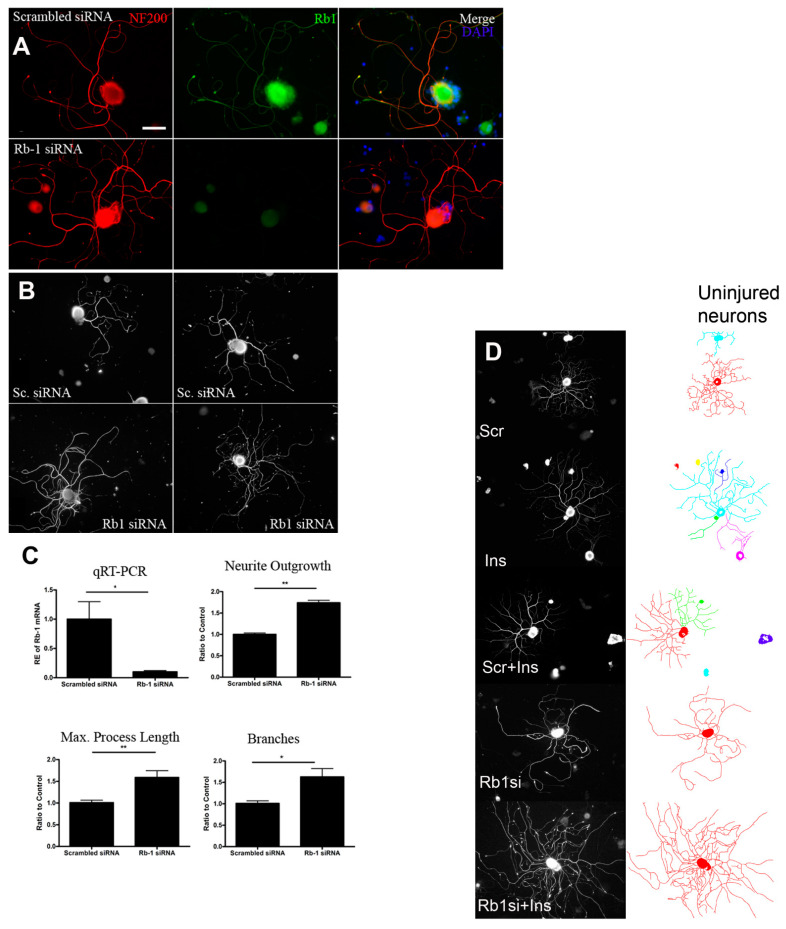
Rb1 is co-expressed with NF200 in injured and dissociated adult DRG neurons (**A**,**B**) and its knockdown by siRNA decreases Rb1 expression compared to a scrambled control siRNA. Rb1 siRNA is associated with rises in normalized neurite outgrowth of adult sensory neurons in culture with quantitation of knockdown and of outgrowth given in (**C**). Scale bar = 50 µM. * and ** *p* < 0.05 and 0.01 respectively. The images (**A**–**C**) are reproduced with permission from Christie et al. and Springer Nature [175]. In (**D**) is illustrated a synergistic impact of neurite outgrowth in adult peripheral sensory neurons in vitro exposed to (from top down) control scrambled siRNA, insulin at 10 nM, both scrambled control siRNA and insulin or Rb1 siRNA and insulin. Compared to the scrambled control siRNA cultures, the most prominent rises were noted after both Rb1 siRNA and insulin were added together. Scale bar = 100 µM. The Image (**C**) is reproduced with permission from Komirishetty et al. and Elsevier [177].

In separate work [177] we examined whether delayed KD of Rb1 locally after a sciatic nerve crush in mice could impact regrowth that is already well underway (14 days post injury). While Rb1 siRNA did not impact electrophysiological recovery (CMAPs and SNAPs), there were improvements in mechanical and thermal sensation and greater re-innervation into the paw epidermis up to 28 days following injury. Given findings that Rb1 siRNA combined with low, subhypoglycemic and trophic doses of local insulin had striking synergy in enhancing neurite outgrowth, sprouting, and branching in vitro (Figure 5), their combination was studied in vivo. Local application of both Rb1 siRNA and low dose insulin synergistically improved motor electrophysiology, thermal and mechanical sensitivity, repopulation of sural myelinated axons, epidermal reinnervation of the skin and SCG10 DRG expression, a marker of regeneration.

### 5.3. APC (Adenomatous Polyposis Coli)-β Catenin

APC is a tumour suppressor protein that is mutated in colorectal tumours. Following activation of the transmembrane Wnt (Wingless-Int-1) signaling pathway described above, APC forms a ‘destruction complex’ with the protein β-Catenin, causing it to be phosphorylated and inactivated. We studied what the impact of this pathway might be in adult sensory neuron regeneration [141]. Both APC and β-Catenin were expressed and colocalized in adult sensory neurons. APC was also expressed in SCs and β-Catenin in perineuronal satellite glial cells. After axotomy however, there was upregulation of APC in neurons and downregulation of β-Catenin. Resembling PTEN, APC was more intensely expressed in slower growing IB4 sensory neurons. Dissociated pre-injured adult rat DRG neurons exhibited greater neurite outgrowth and branching during APC siRNA knockdown, an impact abrogated by concurrent treatment with ICG-001, a β-Catenin inhibitor (Figure 6). Naïve uninjured neurons also exhibited greater outgrowth after APC siRNA. Concurrently, APC siRNA increased β-Catenin expression in DRG neuron nuclei as well as expression of TCF-1 and LEF-1. The findings indicated an elegant repositioning of β-Catenin localization and activity to nuclei following injury. Finally, in mice with sciatic crush and given ipsilateral APC siRNA, there were improvements in mechanical and thermal sensation, conduction velocities and myelinated axon numbers in the sural nerve compared to controls at 28 days following injury.

Taken together, the cases of both E2F1 signaling enabled by Rb1 KD and β-Catenin signaling by APC KD, illustrates an impact on growth through facilitation of transcriptional activity. What is not clear is what the range of activated growth signals from this route are. Moreover, how this transcriptional activation overlaps with well established growth signaling cascades, such as Ras/ERK and PI3K-pAkt is unexplored.

### 5.4. RhoA and Growth Cones

RhoA, Rac1 and CDC42 are growth cone proteins of the Ras superfamily of GTPases that influence cell polarity. While Rac1 and CDC 42 facilitate growth cone advancement through lamellipodia and filopodia, RhoA signals growth cone collapse through Rho kinase (ROK aka ROCK). RhoA activation inhibits spinal cord regeneration [178]. In the PNS, RhoA is activated by extracellular Nogo-66, myelin-associated glycoprotein and chondroitin sulphate proteogycans (CSPGs) [179,180,181,182]. RhoA/ROK activates LIM kinase that phosphorylates and inactivates cofilin, impairing actin turnover but also phosphorylates myosin resulting in growth cone retraction. RhoA and ROK are expressed in DRG sensory neuron cytoplasm and in their distal growth cones [183] with rises peaking at about 4 days following axotomy. Moreover, activated GTP bound RhoA increased in proximal axons and DRGs at 3- and 7-days post axotomy. Pharmacological inhibition of ROK using HA-1077 generated a dose dependent rise in neurite outgrowth of adult DRG sensory neurons whereas local infusion of HA-1077 into regeneration conduits, as described above, increased the numbers and extent of axon outgrowth from the proximal stump of a transected sciatic nerve. Regenerative benefits from the ROK inhibitor fasudil have also been noted by Hiraga et al. with improved motor electrophysiology and increased numbers of regenerating myelinated axons [184]. Joshi et al. suggested that motor neurons might be more responsive to the ROK inhibitor Y-27632b [185]. Overall, the findings identify a local inhibitory mechanism to shut down growth cones and inhibit regeneration in adult axons. However, it is uncertain whether this system might have supraregulatory input from more generalized growth signaling pathways in peripheral neurons. Unlike the plasticity proteins discussed earlier, this roadblock to regrowth appears primarily triggered by the microenvironment of the regenerative milieu.

## 6. Regeneration in Neuropathies, Diabetes

Regeneration of peripheral axons is usually considered in the context of trauma with crush, compression, stretch or transection of nerve trunks. However, despite the ubiquity of traumatic peripheral nerve injuries, damage to nerves in neuropathy is yet more common. The level and extent of axonal damage varies with the type of disorder. For example, Guillain-Barre syndrome, or polyneuropathy (GBS), is classified most commonly as a demyelinating disorder characterized by remyelination and recovery over weeks-months. However, there is a subset of GBS patients in whom primary axonal damage is characteristic and in severe demyelinating versions of GBS, concurrent axonal damage is also common [186]. The latter has been described as a ‘bystander’ effect, with presumed axonal damage as a bystander to nearby zones of inflammation within the nerve. Similarly, among the broad categories of inherited polyneuropathies known as Charcot-Marie-Tooth (CMT) disease, some are predominantly demyelinating and others chiefly axonal in character depending on the genetic abnormality. However, even among the demyelinating CMT neuropathies, chronic loss of motor axons is also a prominent feature, resulting in wasting and weakness of distal extremities. Acute and chronic primary axonal disorders are common and range from ischemic vasculopathies to chemotherapeutic toxic neuropathies, critical illness polyneuropathy and others. Diabetic polyneuropathy is largely an axonal disorder with a degree of superimposed demyelination. Focal or localized neuropathies with axonal damage are also quite common, including severe forms of carpal tunnel syndrome, ulnar neuropathy at the elbow, plexopathies, root damage from disc herniation and many others. While a number of these neuropathies are potentially treatable with immune modulation, decompression and other approaches, their prognosis for recovery depends on the extent of axonal damage. For example, some GBS survivors have remained severely and permanently paralyzed from widespread axonal damage. Irrespective of cause, axonal regeneration is required for recovery. Unfortunately however, no specific therapeutic interventions are yet available to improve or enhance regeneration in neuropathies.

Unlike traumatic nerve injuries, axon damage in neuropathies is not usually associated with nerve trunk disruption, a feature often associated with trauma. SCs, basement membrane, vascular supply and other components of the nerve are usually (not always) intact and available to guide regrowing axons. Despite this support, many persons with axonal polyneuropathies or focal neuropathies do not recover their neurological deficits, an outcome indicating ongoing failure of axon plasticity. This may also be linked to with impaired SC support of partnering axons to regenerate.

Despite an overall neuronal reluctance to regrow, distal epidermal axons within the skin have considerable plasticity, even in the absence of neuropathy or nerve injury [187]. Epidermal axons within intact skin of humans express growth molecules such as GAP43, Shh, SCG10 and others, some locally elaborated by intra-axonal synthesis [167,188]. This ongoing ‘growth state’ or plasticity may enable skin axons to keep apace with normal keratinocyte migration and shedding from the skin. It is also well known that sensory deficits can recover from collateral reinnervation whereby residual intact sensory neurons extend branches into recently denervated targets, such as skin. Diamond and colleagues have shown that the neurotrophic requirements, namely NGF, for collateral reinnervation differed from those of regeneration growth [189]. Given the plasticity of skin innervation, it is possible that local support of regrowth in some sensory neuropathies may offer an alternative route to recovery. For example, an approach to support epidermal regrowth locally through the trophic actions of an M1 muscarinic agonist is currently being trialed in diabetic and other neuropathies [190].

Diabetic polyneuropathy (DPN) is a candidate for regenerative translational solutions, given the absence of current therapy to reverse or arrest the disorder (see recent reviews [191,192,193]). Over half of both type 1 and 2 diabetic persons develop polyneuropathy. As a unique neurodegenerative disorder, sensory neurons are targeted prominently, with retraction of sensory axons from the skin resulting in ‘glove and stocking’ loss of sensation. While several seminal trials of growth factors have not been successful, treatments designed to treat nonspecific oxidative stress, polyol flux, or other common features of diabetic complications have also been unimpressive. In experimental DPN there is evidence that insulin has neuronal trophic properties independent of its glycemic actions, and can reverse key features of the disorder. Insulin receptors are expressed on neurons and glial cells [194,195]. Low dose insulin administered intrathecally or intranasally to access DRGs, near nerve or applied to footpads have all demonstrated impacts on DPN without alterations in blood glucose levels [196,197,198,199,200]. Recent work also indicates that insulin support of sensory neurons may be synergistic with other intrinsic mechanisms to support regrowth, such as Rb1 knockdown [177].

In diabetes mellitus (DM) the PNS experiences a ‘double hit’ from degenerative neuropathy involving loss of distal terminals, such as those innervating skin, but also failure of regeneration [201,202,203]. This is clinically important given the susceptibility of diabetic persons to superimposed focal neuropathies such as carpal tunnel syndrome. While mechanisms for diabetic regenerative failure have been uncertain, considerations have included glycosylation abnormalities of basement membranes, failure of SC support for regrowth, local microangiopathy, attenuated RAG expression, radical stress, polyol flux and others. Singh et al. [204] from our laboratory examined PTEN expression and function in sensory neurons from diabetic mouse models. Chronic type 1 and 2 models had substantial rises in PTEN transcripts and protein in DRGs with heightened levels retained following a superimposed axon injury. PTEN knockdown was associated with increased neurite outgrowth of dissociated adult sensory neurons from diabetic mice. In further work, in vivo application of a PTEN siRNA applied near nerve following injury was retrogradely transported to the ipsilateral DRG where it knocked down PTEN expression. PTEN siRNA was associated with more rapid electrophysiological, behavioural and structural evidence of regeneration-improvements in both motor and sensory conduction velocities, mechanical sensation, myelinated axon repopulation of the tibial nerve and epidermal innervation by 28 days following a sciatic crush injury. The findings provided evidence that PTEN may be an important contributor to nerve regenerative failure in DM and a therapeutic target for translation. Similar findings have been identified in a type 2 model of experimental diabetic neuropathy [205].

## 7. Conclusions

Recovery of motor, sensory and autonomic function following peripheral nerve injury or disease is incomplete following trauma or disease but no specific interventions to enhance regeneration are yet available. While axon regeneration in the PNS exceeds that of the CNS, there are extrinsic and intrinsic barriers to better growth. As a result, longstanding neurological deficits are common and debilitating from nerve trauma or neuropathy. An expanding list of potent extrinsic growth factors has been discovered over the past several decades and their actions have informed us that adult axons are not ‘hard wired’ or static ensembles, but instead capable of considerable plasticity. More recently we are learning that downstream of growth factor signaling involves a series of intrinsic ‘growth pathways’ that can be manipulated in order to foster regenerative growth. Some of these include well explored classical signals such as Ras/ERK and PI3K-pAkt but others involve transcriptional activation. How they overlap and connect is unclear. Among the latter are tumour suppressor molecules, paradoxically expressed in stable regeneration-reluctant neurons and not necessarily classified as a ‘regeneration associated genes’ (RAGs). Their short term and localized manipulation, designed to avoid chronic oncogenicity, enhance neuron plasticity and improve regeneration. Finally local signals that enhance or repress advancement of growth cones may have a role in enhancing adult axon regeneration. Overall, it may be that a combination of approaches have clinical applications to foster regeneration in adult nerve injury or disease.

## Figures and Tables

**Figure 1 ijms-23-13566-f001:**
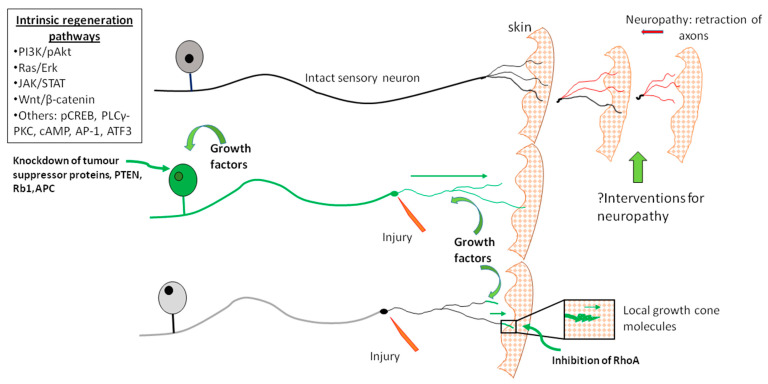
Overall summary of regeneration issues and targets discussed in this review.

**Figure 2 ijms-23-13566-f002:**
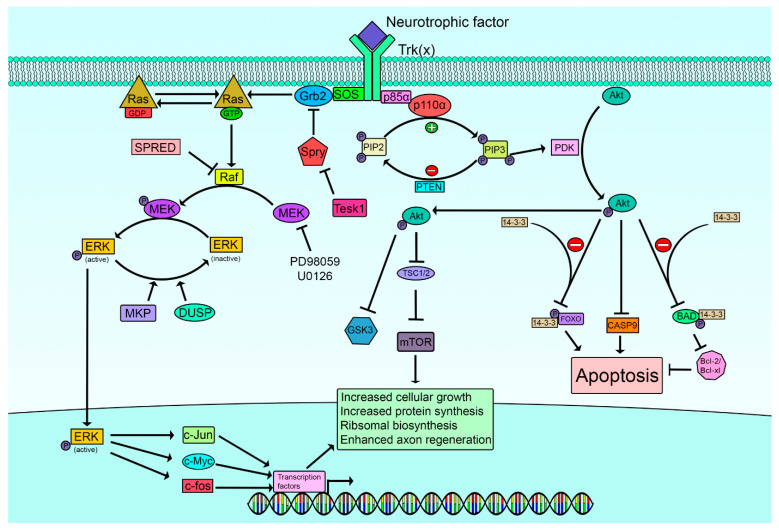
Overview and simplified schemata of the major molecular pathways downstream of neurotrophin growth factors, binding to their Trk receptors, that converge on transcriptional activation.

**Figure 3 ijms-23-13566-f003:**
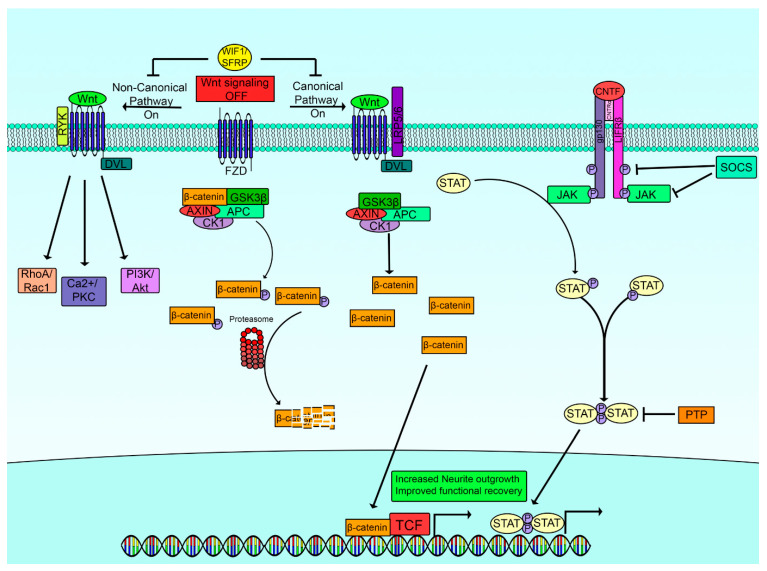
Overview and simplified schemata of major downstream activation pathways involving Wnt/β-catenin and JAK/STAT with their convergence on transcriptional activation.

**Figure 4 ijms-23-13566-f004:**
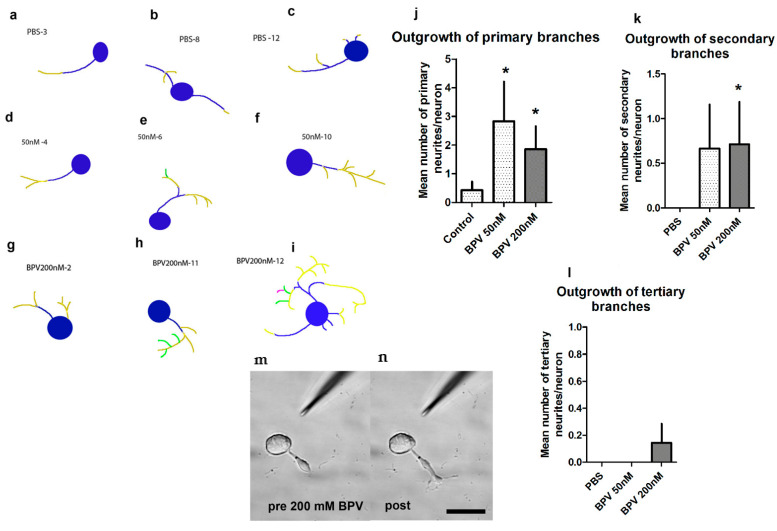
Using a classical growth cone turning assay, perikarya-directed PTEN inhibition induced distal neurite branching. (**a**–**i**) The PTEN inhibitor bp(v)pic (BPV) was directed by a picospritzer to the perikarya, instead of the growth cone, of immunoselected GFRα1 adult sensory neurons. Different doses of BPV ((**d**–**f**) 50 nM BPV and (**g**–**i**) 200 nM BPV) were compared with carrier PBS. The original configuration of the neurons is illustrated in blue. New primary branches are illustrated in yellow, new secondary branches are illustrated in green, and tertiary branches are illustrated in pink. There were significant rises in primary branches (**j**) and secondary branches (**k**), and, in one case, tertiary branches (**l**) when perikarya were exposed to higher dose BPV. An individual neuron imaged by brightfield microscopy before (**m**) and after (**n**) 60 min of growth is illustrated in response to 200 nmol/L BPV applied to its perikarya. Note the rise in branches in neurites distant from the side of application of the PTEN inhibitor. Scale bar = 50 µm.* *p* < 0.05 compared to control. The Figure is reproduced with permission from Guo et al. [172].

**Figure 6 ijms-23-13566-f006:**
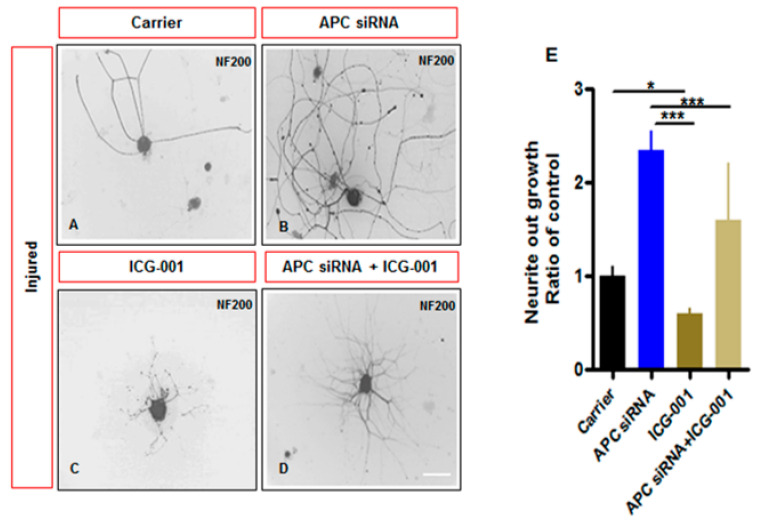
APC participates in the degradation of the signaling molecule β-Catenin. Knockdown of APC using siRNA was associated with increased outgrowth of adult DRG sensory neurons in vitro (**A**–**E**). ICG-001, a selective inhibitor of β-catenin attenuated the impact of APC siRNA on neurite outgrowth. APC knockdown by siRNA was also associated with repositioning of β-Catenin to neuron nuclei (**F**, arrow). Scale bar = 50 µM. * *p* < 0.05, *** *p* < 0.001. The Figure is reproduced with permission from Duraikannu et al. and Springer Nature [141].

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
