# Peer review of "Unleashing Intrinsic Growth Pathways in Regenerating Peripheral Neurons"

_ijms, 2022, doi:10.3390/ijms232113566_

Round 1
Reviewer 1 Report
Although stretched for time and behind the deadline in my review (my apologies), I was pleased to read "Unleashing intrinsic growth pathways in regenerating peripheral neurons" by Trevor Poitras and Douglas W. Zochodne. In covering the field it is no doubt that the authors are well versed in all the findings included in this review. Moreover, I appreciated the commentary provided in section 5. It is refreshing to have authors speak not only the benefits of given techniques but also the limitations. Overall, I thoroughly enjoyed the review and believe it is quite a comprehensive work that helps other researchers in the field. The following are my minor comments to address prior to publication.
1. It is clear that not only are the authors familiar with the field but they have also published a good deal in the field. Overall the findings in the field have been summarized until we get to sections 5.1, 5.2 and 5.3, where their own findings supported with figures are drawn out a bit longer than necessary for readers. I would suggest shortening this part to read the highlights and not so detailed on all the results and also if figures must be shown that they are modified for only the most important sections or made into a schematic. These sections stand in contrast to the incredibly well-written and summarized sections that proceeded.
2. If point one is addressed then there would be space to end better than this review has. Given how well the authors know the field I would like to hear where they believe the field should go next. The current ending basically leaves you abruptly finishing. Although I greatly enjoyed addressing neuroregeneration in neuropathies at the end, it still needs a better conclusion.
3. The first time you mention Trk it is not defined, please do so. I am sure it is an oversite because every other abbreviation is defined.
4. At the end of page 4 did you mean "low pK" or "low pKa"? I assume you just forgot the "a" on the end.
5. Make sure to stay consistent with PI3K or PI3k, sometimes you switch between them.
6. There are some spacing errors throughout, but it is an easy fix that likely came from editing the manuscript prior to submission.
7. Page 17 (727) I think you meant to write "unleashed" instead of "unleased". The next line has an extra "be" that is not necessary.
8. It may just be me, but others may run into the same issue, therefore you may want to mention it as "ROK, aka ROCK", as I had to look it up in case it was different than what I knew.
Overall I do believe these to be minor issues and could be dealt with rather quickly. I look forward to having this review out in publication after these issues are addressed.
Author Response
Thank you for your input. The paper is revised along the lines suggested, all of which were helpful commentaries:
- We agree this section was too wordy, apologies. The sections mentioned were extensively edited and shortened for clarity. We hope this works better. We have reduced the size of Figure 4 but did retain the Figures otherwise as we feel they convey important points we wanted to make. Schematics are provided in Figures 1-3.
- The 'Regenerative Brakes' section is now reduced by 519 words. A 237 word 'Conclusions' section is now added-we hope this helps and agree now that this would have been important to add. The overall Ms, exclusive of Abstract, Figure Legends and references is now 10,263 words, shorter than the previous 10,612 words. We would of course be willing to shorten further if the editor and referees thought it essential to do so.
- Thank you this is defined.
- This was supposed to mean PK, pharmacokinetics but we have reworded to eliminate the concern.
- Thank you, this is corrected.
- We have worked to catch all of the extra spaces.
- This is corrected, sorry.
- We have added the other appropriate abbreviation as suggested.
Reviewer 2 Report
Poitras and Zochodne provide an extensive review on the role of different growth pathways in regenerating sensory neurons. The review is really detailed covering different pathways and molecular players, as well as what occurs in different neuropathies. In general, it seems a really good review of the literature and studies published to date. Despite being quite long, it provides detailed information in specific sections that constitute a good review for researchers interested in the field.
I believe no significant changes are needed but a couple of comments are provided to help improving the manuscript:
Despite the large number of references (>200) cited, no references are provided in section 2 and in a significant part of section 3. Maybe, some could be added to support authors statements.
Figure 4 seems fine (reproduced from a reference, with permission) but appears really large (one full page). It could be reduced significantly.
A summary or conclusions section could be added at the end of the manuscript summarizing the actual situation and the future directions in the regeneration field.
Author Response
Thank you for the helpful input. We have revised the piece along the suggested lines:
- Citations are added to Section 2, and 3 at the beginning of these sections.
- Figure 4 is reduced in size
- A Conclusions section is added.